# Generalization by Specialization: Unveiling Domain Specialized Subnetworks in Generalized Large Language Models

## Abstract

In recent years, large language models (LLMs) have exhibited remarkable generalization capabilities. Previous studies have largely focused on examining the generalization mechanisms in smaller models to draw inferences about similar mechanisms in larger language models. However, these smaller models typically possess limited generalization capacity. In this study, we explore the generalization mechanisms of billion-parameter language models, with a particular attention on publicly available models such as LLaMA and Gemma. Our findings reveal that weight activations exhibit task-specific behavior, indicating that not all weights are necessary for task performance. Building on this insight, we introduce a parameter probing method to identify subnetworks optimized for specific tasks without extensive fine-tuning. This method involves sorting and grouping weight activations followed by the pruning of less significant groups based on a small validation set. Furthermore, our results show that subnetworks specialized for domain-specific tasks achieve improved performance and generalization within their respective domains, but their performance deteriorates across different domains. This study presents a novel perspective on generalization of LLMs where the strength of large language models lies in their multiplicity of domain-specific subnetworks, allowing them to excel in various in-domain tasks.

## 1 Introduction

Large language models (LLMs) have achieved excellent performance on multiple tasks, demonstrating impressive generalization capacity in practical scenarios. Recently, there has been growing interest in the generalization of LLMs (Bayazit et al., 2023; Choenni et al., 2023; Sun et al., 2024). Bayazit et al. (2023) indicated that knowledge-critical subnetworks are contained in pretrained language models. Bhaskar et al. (2024) discovered that a specific set of attention heads in language models act as the core part that ensures generalization by interacting with other attention heads.

However, current methods mainly focus on the small scale of language models, such as BERT (Devlin, 2018) or GPT-2 (Radford et al., 2019). The study on the generalization of large language models, such as LLaMA (Touvron et al., 2023), and Gemma (Team et al., 2024), remains underexplored despite numerous fine-tuning experiments conducted in previous studies.

In this work, we investigate the larger LLMs directly, especially the publicly available LLaMA and Gemma models. We observe that weight parameters in LLMs act consistently when dealing with the same tasks but exhibit different behaviors across diverse tasks. Specifically, weight activations are more similar for analogous questions than for different tasks, suggesting a robust domain-specific mechanism within the generalized LLM architecture. In Figure 2, we observe that the weights exhibit an overlapping pattern when performing the same task. However, these weights tend to diverge when considering different tasks. This raises a crucial question:

*Are all weights necessary for inference on a given task?*

Previous pruning methods (Sun et al., 2024; Frantar & Alistarh, 2023) have demonstrated that LLMs can maintain comparable performance even after portions of the weights are removed. However,

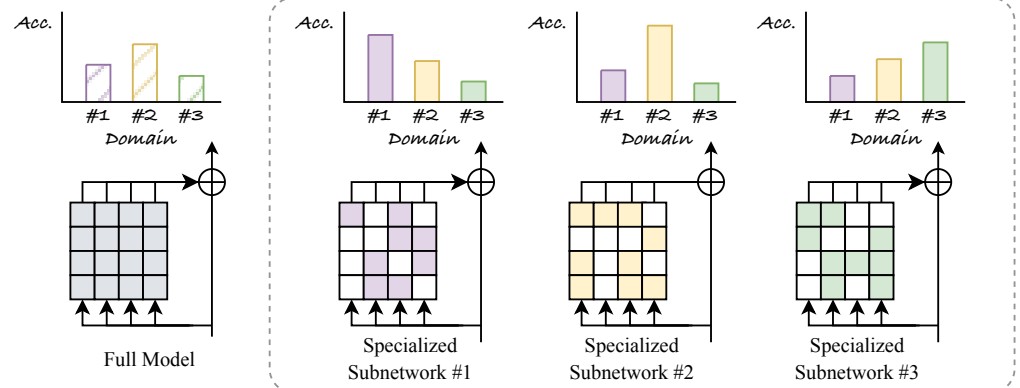

Figure 1: We found that *specialized* subnetworks within a large language model perform *better* than the *full* model for each domain. In addition, these *specialized* subnetworks yield *inferior* performance on other tasks compared to the *full* model. Redundant weights in one domain are always present in the *full* model.

these methods do not guarantee task-specific performance and often require fine-tuning on specific corpora, thus limiting the exploration of LLMs' generalization capabilities across different tasks. In this study, we propose a novel parameter search method that identifies domain specialized subnetworks for given tasks. Our method only requires inference on a few instances for a given task, making it feasible for LLMs with billions of parameters.

Specifically, we group domain-specialized weight neurons based on their scores and analyze their activations across different tasks. Weight neurons highly overlap within the same task, especially in the last layer, while showing significant divergence across different tasks. This consistent activation suggests the presence of specialized subnetworks optimized for specific domains. By pruning the group of weight neurons and evaluating their performance, we uncover domain-specialized subnetworks within a single model. These subnetworks consistently enhance performance across different large language models on the in-domain tasks.

We further examine the generalization capabilities of these specialized subnetworks across different domains. The results highlight a trade-off: *domain-specific subnetworks excel within their respective contexts but show reduced generalization to unrelated tasks*. The specialized subnetworks outperform the full model on their respective domains but underperform on other domains, as seen in Figure 1. Removing redundant weights in the target domain improved performance on related tasks. However, we also discovery that redundant weights in one domain can be significant for other domains, limiting the generalization capacity of these specialized subnetworks.

In summary, our findings enhance the understanding of the generalization mechanisms of LLMs. They underscore the importance of having various specialized subnetworks to ensure model performance in each specialized domain. By ensembling all these specialized subnetworks, the generalization capacity of the LLMs can be achieved.

## 2 RELATED WORK

**Network Pruning.** Pruning is an effective method for network compression that removes layers or parameters to produce sparse networks (LeCun et al., 1989), which can be roughly divided into structured and unstructured schemes. Structured pruning, also known as activation pruning, removing redundant channels or filters to decrease computational complexity and memory demands while maintaining the overall network architecture. To be specific, Babaeizadeh et al. (2016) develops a fully automated pruning algorithm leveraging the correlation between neuron activations in hidden layers. Dubey et al. (2018) employs coreset representations in network pruning, eliminating unnecessary weights and neuronal activations within CNNs for compression. Recently, there has been a surge in integrating LLMs into pruning (Ma et al., 2023; Bansal et al., 2022; Voita et al., 2023; Liu et al., 2023; Sun et al., 2024; Bhaskar et al., 2024), which demonstrates the prompt-dependent

and task-specific sparsity in LLM structural components (*e.g.*, attention heads and MLP neurons). Unlike structured pruning, unstructured pruning removes weights without considering the overall architecture, targeting specific weights or neurons on a finer granular level (Han et al., 2015; Hoang & Liu, 2023; Paul et al., 2022; Sun et al., 2024). In this paper, we mostly utilize unstructured pruning based on Sun et al. (2024) to eliminate redundant weights.

**Subnetworks.** A group of representative studies evidence that neural networks are composed of modular subnetworks, each responsible for specific subtasks (Lepori et al., 2023; Choenni et al., 2023; Hupkes et al., 2020). Concretely, Nooralahzadeh & Sennrich (2023) introduces task-specific subnetworks to bridge the gap in cross-lingual transfer. Similarly, Choenni et al. (2023) proposes language-specific subnetworks that guide selective parameter sharing in multilingual training to enhance positive transfer. Bayazit et al. (2023) discovers a sparse subnetwork within GPT-2 that in charge of specific collections of relation knowledge, with no performance drop when it is removed. Previous work manages to localize the knowledge-critical weights in LMs. In this work, we find redundant weights that are not necessary for input tasks. By removing the redundant weights, the sparse model achieves higher in-domain performance.

**Generalization.** The generalization of neural networks has been a long-standing research in NLP community. McCoy et al. (2019) reveals that fine-tuning BERT with different random seeds results in varying generalization, despite achieving similar accuracy on in-domain tasks. Another line of efforts delve into out-of-distribution setting, which finds that performance consistently improves even when in-domain performance appears to be saturated. A striking phenomenon, "Grokking", is recently proposed by Pearce et al. (2023). When training a set of samll models on toy tasks, the models can process and understand unseen data after an extended training phase, effectively leveraging the accumulated knowledge.

## 3 PROBLEM SETUP

**Dataset.** In this work, we focus on several popular benchmarks that evaluate the generalization capacity the large language models.

- **MMLU** (Hendrycks et al., 2020) is a large-scale multi-task dataset consisting of 57 tasks spanning a wide range of subjects, including elementary mathematics, US history, computer science, and law. It requires models exhibit a broad knowledge base and proficient problem-solving skills.
- **GSM8K** (Cobbe et al., 2021) is a dataset containing $8,500$ grade school mathematics questions along with their corresponding natural language solutions. It is used to probe the informal reasoning ability of large language models.
- **MBPP** (Austin et al., 2021) is a programming benchmark with $1,000$ crowd-sourced Python problems, in which $974$ tasks solvable by entry-level programmers. It is formulated to assess the ability of models in generating short Python programs.
- **HumanEval** (Chen et al., 2021) is a collection of $164$ authentic programming challenges implemented in Python, with each generation task involving an average of 7.7 test cases.

**Model.** We mainly employ LLaMA (Touvron et al., 2023) and Gemma (Team et al., 2024) for experimental analysis. Specifically, LLaMA2-7B is a generative text model comprising 7 billion parameters. It has been pre-trained and fine-tuned on a dataset of 2 trillion tokens. Gemma-7B is a lightweight and text-to-text decoder-only large language model, developed with the same underlying principles as the Gemini model (Team et al., 2023). It has been trained on a diverse array of text sources, collectively amassing a total of 6 trillion tokens.

**Weight Score.** In language models, the input $X$ to each linear layer is a tensor with a shape of $BL \times C_{in}$, where $B$ and $L$ denote the batch size and the length of the sequence respectively. Let $W \in \mathbb{R}^{C_{in} \times C_{out}}$ be the weight parameter in the linear layer. Each neuron in the weight matrix would calculate with every token in inputs. To ascertain the impact of each weight neuron, we follow Sun et al. (2024) for calculating the score:

$$S_{i,j} = |W_{ij}| \cdot ||X_j||_p, \tag{1}$$

where $|\cdot|$ represents the absolute value operator, $||X_j||_p$ evaluates the $\ell_p$ norm of $j^{th}$ features aggregated across $B \times L$ different tokens. The final score is computed by multiplying these two scalar

values, with a higher score indicating greater contributions to the outputs. Note that we utilize the $\ell_2$ norm for score calculation. In Sun et al. (2024), pruning the weight neurons with the lowest scores achieves comparable performance to the original LLMs. However, the pruned model always performs lower performance than the full model. The study explores *how the weights affect different tasks and finds the optimal subnetwork tailored for different domain tasks.*

# 4 WEIGHT DISTRIBUTION IN LLMS

## 4.1 GROUPING DOMAIN-SPECIALIZED WEIGHTS

To investigate how the weights LLMs acts to different tasks, we explore the weight scores in LLMs for various tasks. Suppose we have several instances denoted by $\boldsymbol{X}^{t,k} \in \mathbb{S}^t$, where $\mathbb{S}^t$ is the instance pool for the $t^{th}$ task and $k$ is the number of instances. In each layer of the model, we can easily obtain the weight score $\boldsymbol{S}^{t,k}$ based on Equation (1) for the selected instances from the $t^{th}$ task.

*Are the weight activations similar for different instances of the same task?* To analyze the weight distribution across different domain inputs, we sort each row of the weight score matrix $\boldsymbol{S}_i^{t,k}$ and save the corresponding index of weight scores:

$$\boldsymbol{I}_i^{t,k} = \{\sigma(1), \sigma(2), \ldots, \sigma(n) \mid m < n \Rightarrow \boldsymbol{S}_{i,\sigma(m)}^{t,k} \geq \boldsymbol{S}_{i,\sigma(n)}^{t,k}\}, \tag{2}$$

where $\sigma(\cdot)$ is the ranking function to sort the weight scores in the descending order. We then split the indices of weight scores into $G$ groups to investigate the distribution of weight scores for different tasks. The $j^{th}$ grouped indices is saved by:

$$\boldsymbol{G}_{i,j}^{t,k} = \{\sigma(j \cdot v), \cdots, \sigma(j * v + v)\}, \tag{3}$$

where $v = \frac{C_{out}}{G}$ is the size of the $j^{th}$ group and $\bigcup_{j=1}^g \boldsymbol{G}_{i,j}^{t,k} = \boldsymbol{I}_i^{t,k}$. The first group contains the weight indices that corresponds to weight neurons achieved highest weight scores. In each layer, we can obtain the distribution of weight scores. Let $\boldsymbol{X}^{t_1,k}$ and $\boldsymbol{X}^{t_2,k}$ denote the number of $k$ instances sampled from the domain tasks $t_1$ and $t_2$ separately. To compare the difference of weight distribution across same and different tasks, we calculate the overlap between weight indices within the same group:

$$\texttt{Overlap}(\boldsymbol{X}^{t_1,k}, \boldsymbol{X}^{t_2,k})_j = \frac{1}{C_{in}} \sum_i |\boldsymbol{G}_{i,j}^{t_1,k} \cap \boldsymbol{G}_{i,j}^{t_2,k}| / |\boldsymbol{G}_{i,j}^{t_1,k} \cup \boldsymbol{G}_{i,j}^{t_2,k}|. \tag{4}$$

The higher overlap value indicates that weight neurons are consistent for the two inputs. Similarly, we can calculate the overlap between weight indices of the same task that contains different instances. In addition, we calculate the accumulated overlap of weight indices between two tasks:

$$\texttt{Overlap\_accum}(\boldsymbol{X}^{t_1,k}, \boldsymbol{X}^{t_2,k})_m = \frac{1}{C_{in}} \sum_i |\bigcup_{j=1}^m \boldsymbol{G}_{i,j}^{t_1,k} \cap \bigcup_{j=1}^m \boldsymbol{G}_{i,j}^{t_2,k}| / |\bigcup_{j=1}^m \boldsymbol{G}_{i,j}^{t_1,k} \cup \bigcup_{j=1}^m \boldsymbol{G}_{i,j}^{t_2,k}|. \tag{5}$$

## 4.2 COMPARISON BETWEEN DOMAIN WEIGHTS

To study the effect of the weight neurons on different domain tasks, we select multiple instances from distinct domains. These samples are feed into the network and we obtain the weight scores for comparison. In this paper, we mainly compare the weight distribution of different tasks based on Equation (4) and Equation (5).

We begin by selecting a base domain and comparing its weight distribution with that of other domains. To illustrate the difference in weight distribution within a domain, we randomly sample varying numbers of instances from the same task and calculate the weight overlap. For example, we

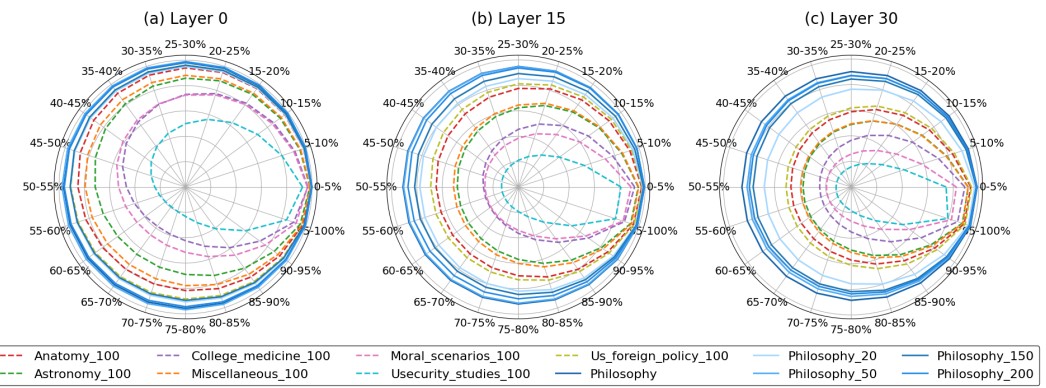

Figure 2: **The weight overlap between *Philosophy* domain and other domains on different layers**. The weights are divided into 20 groups across all three layers. A blue solid line represents weight overlap within the same tasks, while a dashed line represents comparison between different tasks.

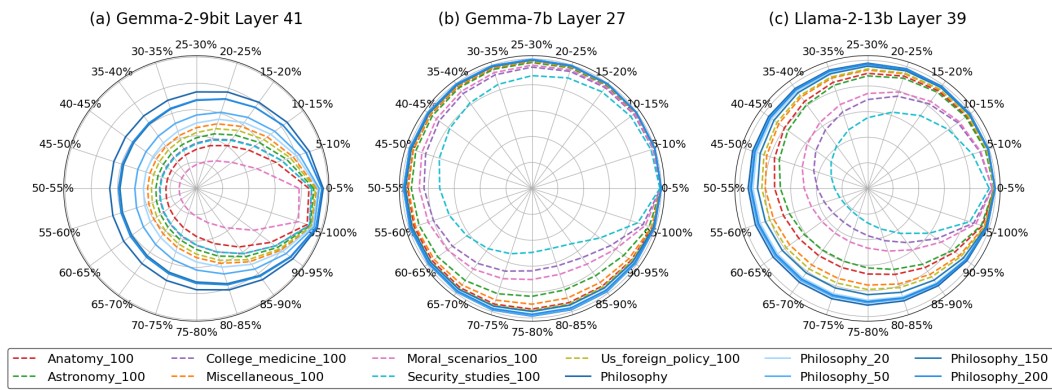

Figure 3: **The weight overlap between *Philosophy* domain and other domains on the last layer of different large language models**. A blue solid line represents weight overlap within the same tasks, while a dashed line represents comparison between different tasks.

randomly select 20, 50, 100, 150, and all instances related to the *philosophy* task. For other tasks, we randomly select 100 instances to calculate the grouped indices accordingly. The paper primarily explores weight neurons in terms of **layer**, **task**, **model**, and **weight group** to demonstrate the similarity among different weight groups.

**Layer.** The public LLaMA2-7B model consists of 32 transformer layers, each of which has multiple fully connected layers. We investigate the weight distribution on different fully connect layers, and select *Philosophy* as the base domain task for comparison. The compared domains including 7 categories from MMLU dataset, such as *Antonomy*, *Astronnomy*, *College Medicine*, *European History*, *Miscellaneous*, *Moral Scenarios*, *Security Studies*, and *US Foreign Policy*. For each category, we randomly select 100 instances and calculate the weight scores. The weight neurons are split into 20 groups. The overlap of weight distribution across different tasks are visualized in Figure 2.

Weight neurons in all layers highly overlap when input instances are sampled from the same tasks. Yet, the overlap between instances of the same task is lower in the first layer. Notably, the overlap between instances of different tasks decreases significantly from the first layer to the last layer. Thus, it is easy to observe similar weight neuron patterns in the last layer for inputs from the same task.

Furthermore, an interesting finding is that the overlap of weight neurons is higher in the first and last group. The first group contains indices where the highest weight scores are attained, indicating that critical weights are similar across all tasks. Additionally, we demonstrate that removing weight neurons in the higher ranked group leads to a dramatic decrease in model performance on all tasks.

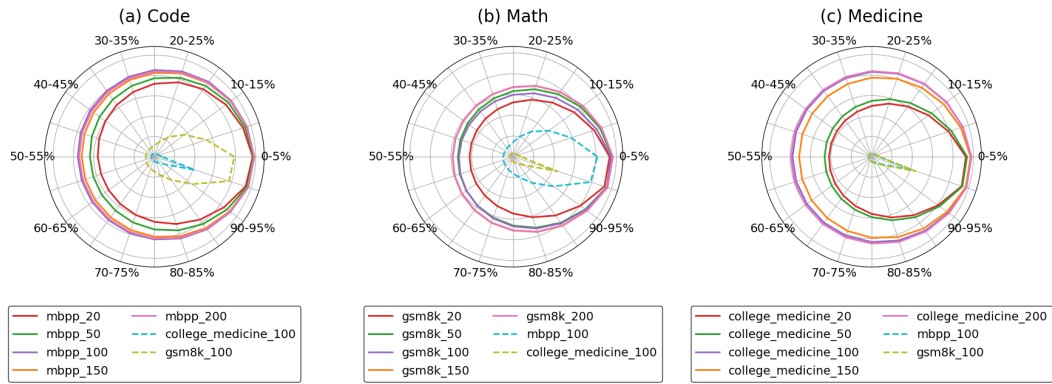

Figure 4: **The weight overlap between the different domains**. For *Code* domain, we sample various number of instances from MBPP dataset (Austin et al., 2021). For *Math* domain, we sample various number of instances from GSM8K dataset (Cobbe et al., 2021). The dashed line represents comparison between different tasks.

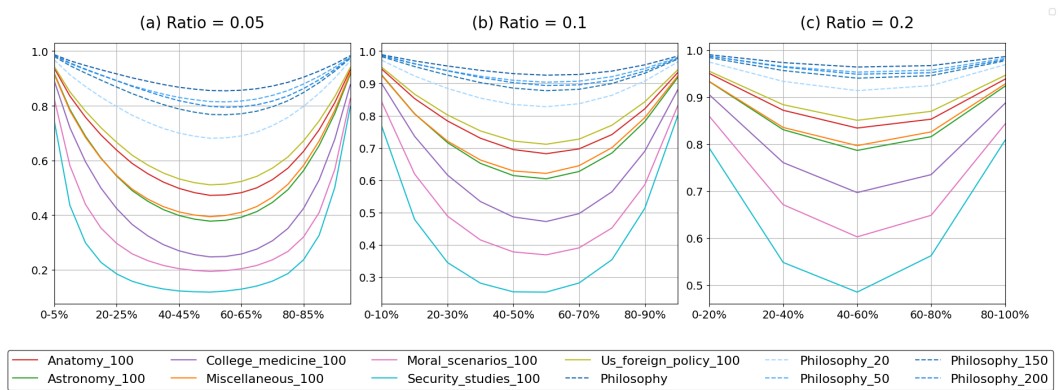

Figure 5: **The weight overlap across different numbers of groups**, with a ratio of 0.05 indicating that the weight neurons are divided into 20 groups. The dashed line illustrates the comparison for similar tasks.

**Model.** To explore if the observed findings are specific to the LLaMA2-7B model or are applicable to models at various scales, we conducted experiments on Gemma2 model series and the LLaMA2-13B model. The results, illustrated in Figure 3, consistently show that weight neurons heavily overlap for instances of the same task, regardless of the input domain. Besides, the critical weights remain similar across different input domains. The weight distribution of LLaMA2-7B and LLaMA2-13B is similar, while the pattern appears different for Gemma-7B and Gemma2-9B.

**Task.** We validate this finding by selecting different base domain tasks such as *Medicine*, *Code*, and *Math*. For *Medicine* domain, instances were sampled from MMLU dataset, while for *Math* domain, instances were sampled from GSM8K dataset, and for *Code* domain, instances were sampled from MBPP dataset. Weight distributions in Figure 4 exhibits a significant overlap when inputs originate from the same domain, suggesting that weight neurons activate in a similar pattern. Notably, there is a minimal weight overlap between the *Medicine* and *Code* domains. This discrepancy could be attributed to the substantial differences in inputs from these domains and the potential variation of critical weights when the domain knowledge greatly diverges. Interestingly, the weight similarity between *Code* and *Math* domains surpasses that of the *Code* and *Medicine* domains, despite the distinctiveness of inputs from all three domains.

**Weight Group.** In previous settings, weight neurons are divided into 20 groups. We examine how the distribution varies across domains with different numbers of weight groups. In Figure 5, we demonstrate that weight overlaps increase with larger group ratios. As group ratio increases,

---

**Algorithm 1** Probing Domain Specialized Subnetwork.

---

**Input:** Instances $\boldsymbol{X}$ sampled from the target domain and the corresponding groundtruth $\boldsymbol{Y}$. The number of weight group $G$. The large language model $f$ and number of layers $L$ in the model. The weight mask $\boldsymbol{M}$ which is initialized as ones with same size of the weights in all layers.

1: Initialize the optimal domain model by $f_p \Leftarrow f$
2: Evaluate the model performance on the given domin by $\ell(f_p(\boldsymbol{X}), Y)$
3: **for** $g \in 1, \ldots, G$ **do**
4:    **for** $l \in 1, \ldots \ldots, L$ **do**
5:       Calculate the group of weight indices $\boldsymbol{G}$ in each layer by Equation (3)
6:       Prune the weight by setting the weight mask $\boldsymbol{M}_{i,j} = 0$ if $j \in \boldsymbol{G}_{i,g}$
7:       Update the weight $\boldsymbol{W}' = \boldsymbol{M} \cdot \boldsymbol{W}$
8:    **end for**
9:    Save the updated model $f_g$ and evaluate the updated model $\ell(f_g(\boldsymbol{X}), Y)$
10:   **if** $\ell(f_g(\boldsymbol{X}), Y) < \ell(f_p(\boldsymbol{X}), Y)$ **then**
11:      Update the subnetwork $f_p \Leftarrow f_g$
12:   **end if**
13: **end for**
14: **Output:** The optimal subnetwork $f_p$ for the input domain

---

the overlap intensifies, making domain differences more evident and reducing variations within the same domain.

After comparison of distribution of weight scores across different domains, we observe the domain principle in generalized LLMs. For different inputs from the same task, the weights in LLMs exhibits highly consistent activation. This consistent activation across inputs within the same task domain points to a robust domain-specific mechanism within the generalized architecture of the LLMs. Such a mechanism suggests that the model has developed specialized subnetworks or pathways that are optimally tuned to process and respond to particular types of linguistic tasks, thereby maximizing efficiency and performance.

This inherent domain principle could be pivotal in explaining the model's generalization ability. When a model encounters a new but related input, the pre-trained domain-specific weights are already finely adjusted to handle the nuances of such tasks, thus facilitating a smooth transfer of learned knowledge. Consequently, ensuring that the language model's training regimen involves a diverse array of tasks and inputs might enhance the development of well-tuned domain-specific pathways, ultimately improving the model's versatility and robustness.

## 5 DOMAIN SPECIALIZED SUBNETWORKS

### 5.1 PROBING DOMAIN SPECIALIZED SUBNETWORKS

In practical applications, leveraging the domain principle improves computational resource utilization. By identifying the activated parts of the model for specific tasks, targeted optimization like pruning redundant pathways and concentrating computational power on relevant subnetworks can be implemented. This enhances processing speed and reduces resource consumption without compromising performance. In this section, we aim to understand the relationship between LLM weights and domain tasks.

We investigate whether our results are due to domain specialized subnetworks. The large language model consists of multiple subnetworks that perform better in specific domains but have less effective generalization compared to the full model. Previously, we grouped weight neurons in each layer based on weight scores for inputs from specific domains. Herein, we further explore the influence of weight neurons at the group level and identify optimal subnetworks for different domains.

To assess the effect of weights on target domains, we evaluate the performance of a pruned subnetwork by removing weight neurons in each group defined as $\boldsymbol{G}^t$ based on Equation (1) for task $t$. For simplicity, we directly set the weight neurons to zero to neutralize their influence on outputs. The output is calculated as follows:

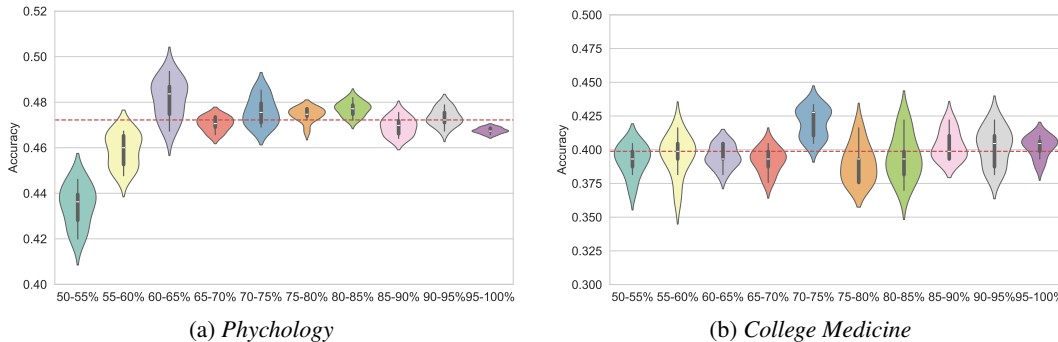

(a) *Phychology*          (b) *College Medicine*

Figure 6: **Pruning the LLaMA-7B** (Touvron et al., 2023) **model at different weight groups**. The dashed line refers to the accuracy of the full model performance.

$$\boldsymbol{X}^{out} = (\boldsymbol{M} \cdot \boldsymbol{W})^{\top} \boldsymbol{X}^{in}, \tag{6}$$

where $\boldsymbol{M}$ is the weight mask to indicate whether the weight neurons is retained. To identify optimal subnetworks for specific domains, we iteratively remove the neurons associated with one group across all layers. From this process, we obtain a set of $G$ subnetworks for inputs from a particular domain task. These subnetworks are then evaluated on matching domain test sets. The top-performing model is saved as the optimal subnetwork for that domain. Further details on domain-specific network probing can be found in Algorithm 1.

Compared to the previous method (Sun et al., 2024) that removes weight neurons with low weight scores, our approach splits weight neurons into groups and eliminates redundant ones based on the performance of specific domain inputs. We find that the contribution of weight neurons to the outputs is not directly indicative of their utility, and weight scores may not accurately reflect the significance of weights. By probing the grouped weights, we achieve a more refined and efficient elimination of weight neurons, resulting in improved overall effectiveness and a targeted and precise solution.

## 5.2 COMPARISON BETWEEN SPECIALIZED SUBNETWORKS

In this section, we compare specialized subnetworks obtained from our method across different domains to examine their performance compared to the full model. Additionally, we explore the variations between specialized subnetworks and the full model, particularly when the base models and domain tasks differ.

**Subnetwork Probing.** In Figure 6, we present the performance of subnetworks in *Phychology* and *Medicine* domains. The experiments are conducted on the LLaMA2-7B model, with 20 groups of weight neurons and each subnetwork being pruned 5%parameters at a time. For each weight group, we randomly sample 30 instances from the validation set in the target domain for 10 iterations. The performance on the test set is reported for each pruned subnetwork.

The red dashed line in Figure 6 represents the performance of the full model. We observe that the optimal subnetworks in each weight group outperform the full model in both *Phychology* and *Medicine* domains, despite removing weights. This observation implies that not all weights are necessary for the target domain. For each task, we can always find a domain-specialized subnetwork that performs better than the full model.

In comparison to subnetworks that prune weights with lower scores, our method achieves higher performance by pruning different groups of weights. Removing the last group of weight neurons does not guarantee performance improvements in the target domain. Conversely, removing the highly ranked group of weight neurons results in a dramatic decrease in performance for each task.

**Redundant Weights.** We conduct experiments on *Anatomy*, *Security*, and *US Policy* domains to investigate redundant weights for different tasks. For each task, we determine the number of weight

| Dataset | Full Model | Pruning at Ratio | | | | | |
|---|---|---|---|---|---|---|---|
| | | 2% | 5% | 10% | 20% | 25% | 50% |
| *Anatomy* | 37.78 | 42.22$_{(\uparrow 4.44)}$ | 42.22$_{(\uparrow 4.44)}$ | 44.44$_{(\uparrow 6.66)}$ | 42.22$_{(\uparrow 4.44)}$ | 42.22$_{(\uparrow 4.44)}$ | 41.48$_{(\uparrow 3.70)}$ |
| *Security Studies* | 52.65 | 53.88$_{(\uparrow 1.23)}$ | 52.65$_{(--)}$ | 52.65$_{(--)}$ | 47.76$_{(\downarrow 4.89)}$ | 43.27$_{(\downarrow 9.38)}$ | 32.65$_{(\downarrow 20.00)}$ |
| *US Foreign Policy* | 68.00 | 68.00$_{(--)}$ | 70.00$_{(\uparrow 2.00)}$ | 70.00$_{(\uparrow 2.00)}$ | 68.00$_{(--)}$ | 68.00$_{(--)}$ | 53.00$_{(\downarrow 15.00)}$ |

Table 1: **Pruning the LLaMA-7B** (Touvron et al., 2023) **at different ratio**. We use ↑ and ↓ to denote improvements and decrements of the specialized subnetwork compared to the full model. The best subnetworks are pruned at varying ratios for different tasks.

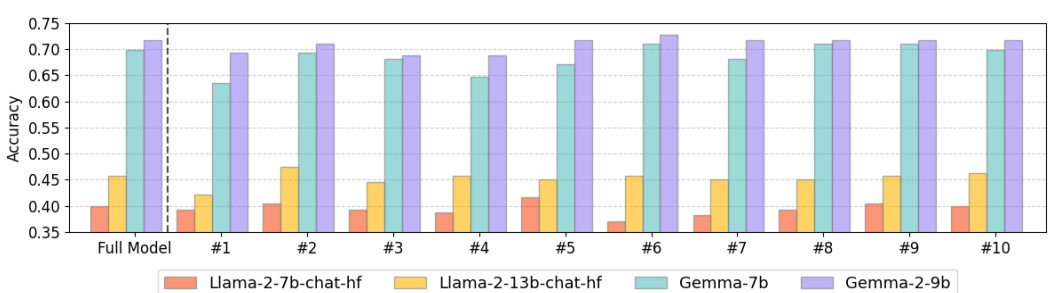

Figure 7: **The specialized subnetwork for *Medicine* with different large language models**. The specialized subnetworks perform better than the full model, even though their performance varies across different language models.

groups and search for the optimal subnetwork. The performance of the best subnetwork for each task at various pruned ratios is presented in Table 1.

Although the input prompts for the present three domains (*i.e.*, *Anatomy*, *Security*, and *US Policy*) are similar as they are all from MMLU dataset, we observe significant differences in redundant weights across these domains. In *Anatomy* domain, even at a 50% pruned ratio, the subnetwork still demonstrates model improvements compared to the full model. The highest performance is achieved when removing 10% of the weights. Conversely, for *Security*, the optimal subnetwork only removes 2% of the weights. Increasing the number of pruned weights results in a significant decrease in model performance for this task.

**Compared Models.** We search for specialized subnetworks in various large language models. In Figure 7, we demonstrate the test performance of pruned weights at different groups for LLaMA2-7B, LLaMA2-13B, Gemma-7B, and Gemma2-9B models. However, in all models, pruning the weights with low weight scores does not result in optimal subnetworks. Nevertheless, by using our method, we identify domain specialized subnetworks that outperform the full model in the target domain for different group indices. This finding indicates that while our method can help find better domain subnetworks, the weight scores may not be the perfect metric for measuring redundant weights in the target domain.

**Speciality *v.s.* Generalization.** After evaluating specialized subnetworks obtained from different domain tasks, we analyze their specialty and generalization performance. Specifically, we focus on four domains: *Medicine*, *Math*, *Code*, and *Philosophy*. Using a 5% pruned ratio, we identify the optimal specialized subnetwork for each domain and assess its performance in both in-domain and out-of-domain tasks. The results in Table 2 demonstrate that all domain specialized subnetworks show significant improvements compared to the full model when applied to in-domain tasks. However, the generalization capacity of these specialized subnetworks tends to decrease. Notably, for *Code* domain, we utilize instances from MBPP dataset as a validation set to select the optimal subnetwork, and then evaluate its performance on the test set of HumanEval dataset. Despite the difference in datasets used for pruning and testing, the *Code*-specialized model still exhibits improved performance. This observation suggests that the improvement achieved by specialized subnetworks in in-domain tasks is not solely dependent on the dataset.

Furthermore, we note that the specialized subnetworks for *Medicine* and *Math* domains demonstrate slightly higher performance in *Philosophy* domain. This could be attributed to the fact that the

| Dataset | Full Model | Specialized Subnetwork | | | |
| --- | --- | --- | --- | --- | --- |
| | | *Medicine* | *Math* | *Code* | *Philosophy* |
| *College Medicine* | 39.88 | 41.04$_{(\uparrow 1.16)}$ | 38.74$_{(\downarrow 1.14)}$ | 38.15$_{(\downarrow 1.73)}$ | 35.26$_{(\downarrow 4.62)}$ |
| *GSM8K* | 18.58 | 15.92$_{(\downarrow 2.66)}$ | 19.47$_{(\uparrow 0.89)}$ | 14.16$_{(\downarrow 4.42)}$ | 13.27$_{(\downarrow 5.31)}$ |
| *HumanEval* | 22.73 | 20.45$_{(\downarrow 2.28)}$ | 22.73$_{(--)}$ | 25.00$_{(\uparrow 2.27)}$ | 18.19$_{(\downarrow 4.54)}$ |
| *Philosophy* | 53.38 | 54.02$_{(\uparrow 0.64)}$ | 54.98$_{(\uparrow 1.60)}$ | 52.73$_{(\downarrow 0.65)}$ | 56.91$_{(\uparrow 3.53)}$ |

Table 2: **The performance of specialized subnetworks within and beyond their domain**. We use ↑ and ↓ to denote improvements and decrements of the specialized subnetwork compared to the full model. The domain specialized subnetwork excels within its field but demonstrates weaker performance outside its domain compared to the full model.

redundant weights specific to *Philosophy* are more numerous compared to the other two domains. Additionally, we calculate the weight overlap between pruned weights of the *Code* and *Medicine* domains, revealing no overlap between the redundant weights of these two domains. However, when pruning the redundant weights of both domains together, the performance on two tasks also decreases. This suggests that redundant weights for one domain can be significant for another domain.

In our experiments, we consistently find specialized subnetworks that excel in in-domain tasks for various LLMs. However, no pruned subnetwork attains the same generalization performance as the full model. By removing redundant weights specific to the target domain, the model achieves good performance on that task. Given the different specialized subnetworks, the redundant weights within a domain may be important for out-of-domain tasks. Our experimental results suggest that the exceptional generalization capacity exhibited by large language models can be attributed to the presence of multiple domain-specialized subnetworks. This implies that any attempts to solely rely on pruned subnetworks, without considering the interconnected and collaborative nature of the language model's subcomponents, may limit its ability to generalize effectively. Therefore, it becomes imperative to strike a balance between specialized subnetworks and the redundancies they possess

## 6 CONCLUSION

In this work, we explored the role of weight neurons in large language models (LLMs) and their specialization across different domain tasks. By analyzing weight scores derived from various tasks, we established that weight neurons exhibit distinct activation patterns tailored to specific domains, suggesting the presence of specialized subnetworks within the generalized LLM architecture. We extended our analysis by probing specialized subnetworks for domains such as Medicine, Code, and Math. Despite achieving improved performance within specialized domains, these pruned subnetworks exhibited reduced generalization capabilities compared to the full model. This observation suggests that the pruned subnetworks, though optimized for specific tasks, may lack the robustness required for broader generalization. In conclusion, our study reveals the inherent complexity and specialization within LLMs, emphasizing the need for balanced training regimens that foster both specialized and general capabilities. Future research could further optimize these domain-specific pathways, releasing the full potential of large language models and bolster their generalization performance.

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

# A APPENDIX

