# OpenReview forum: "Generalization by Specialization: Unveiling Specialized Subnetworks in Large Language Models"
_ICLR.cc/2025/Conference — ICLR 2025 Conference Withdrawn Submission_

### Official Review · Reviewer_wtMy · 2024-10-29

**Soundness:** 3
**Presentation:** 2
**Contribution:** 1
**Rating:** 5
**Confidence:** 5

**Summary:**

The author utilized pruning methods to analyze the internal parameters of the network and concluded that subnetworks exist within the network. The capabilities of LLMs in specific tasks primarily rely on specific subnetworks.

Based on this, the author optimized the network pruning method and proposed a new pruning approach to obtain domain-specific subnetworks, thereby enhancing the network's performance in in-domain tasks.

**Strengths:**

1. This paper conducted experiments on multiple models from the LLama and Gemma series. The results validated the author's hypothesis and previous understanding that multiple subnetworks exist within the model, each representing the model's capabilities in specific domains. These subnetworks collectively form the generalization of the LLMs.

2. The method proposed by the author is very simple, and easy to reproduce.

**Weaknesses:**

1. The domain-specific characteristics within LLMs have been frequently observed in recent research across various fields [1-6]. Notably, [1] utilizes almost the same technique (wanda). Apart from validating this point, the author did not provide other innovative content, which limits the paper's contribution.
2. The experimental results show that in some aspects, the pruned model outperformed the full model, which reduces the credibility of the experimental results.
3. Lack of baseline.

[1] Wang, Yudong, Damai Dai, and Zhifang Sui. "Exploring Activation Patterns of Parameters in Language Models." arXiv preprint arXiv:2405.17799 (2024).

[2] Wang, Lean, et al. "Label words are anchors: An information flow perspective for understanding in-context learning." arXiv preprint arXiv:2305.14160 (2023).

[3] Xia, Mengzhou, et al. "Sheared llama: Accelerating language model pre-training via structured pruning." arXiv preprint arXiv:2310.06694 (2023).

[4] Zhang, Yichi, et al. "PyramidKV: Dynamic KV Cache Compression based on Pyramidal Information Funneling." arXiv preprint arXiv:2406.02069 (2024).

[5] Razdaibiedina, Anastasia, et al. "Progressive prompts: Continual learning for language models." arXiv preprint arXiv:2301.12314 (2023).

[6] Huang, Yufan, et al. "Continual learning for text classification with information disentanglement based regularization." arXiv preprint arXiv:2104.05489 (2021).

**Questions:**

1. Did the author ensure that there is no overlap between the calibration set and the test set? Additionally, can the author provide an explanation of why the pruned model performs better than the original model?

2. In Algorithm 1, does the initial setting of f_p really not affect the algorithm? Does the algorithm prune one group at a time? Is the pruning ratio fixed, and does the algorithm ultimately yield different models pruned from different groups? Or, as the algorithm progresses, the prune ratio changed, attempting to prune different groups in one model. Can the author provide a clearer explanation of the algorithm?

3. Did the author attempt to test the results on LLama3? As far as I know, the capabilities of Llama3 degrade in various aspects after pruning compared to earlier models (Llama2). Does this imply that as models are more fully trained, subnetworks become less distinct, and the model's capabilities become more intertwined within the same weights? Can the author provide more discussion on this?

4. Can the author clarify whether there are any other novel findings in this paper besides validating the existence of domain-specific characteristics within the model? In fact, I believe the author has provided a clearer and more precise analysis of domain-specific characteristics compared to previous work. However, the novelty is limited for ICLR.



Minor comments/questions

1. Many figures in the paper (e.g., Figures 2, 3, 4, 5) do not have well-explained scales. In Figures 2, 3, and 4, it seems that points closer to the center indicate a lower degree of overlap. Does the outer circle represent the range of rankings? In Figure 5, is the ratio equal to 1/group_num? It seems so, but there is no explanation in the paper.
2. Many domains mentioned in Section 5 (e.g., medicine, philosophy) are not clearly explained. Are they from MMLU? These should be mentioned in Section 3.

---

### Official Review · Reviewer_upwi · 2024-11-01

**Soundness:** 3
**Presentation:** 3
**Contribution:** 2
**Rating:** 3
**Confidence:** 4

**Summary:**

This work studies the existence of specialized sub-networks in large networks. It defines a method to divide the network weights in prunable subgroups. It shows that pruning domain-specific weights improves in-domain performance but limits cross-domain generalization.

**Strengths:**

**Diverse topic and tasks**

The study looks at medicine, code, philosophy, etc for benchmarks like MMLU, GSM8K, and HumanEval. I also appreciate the cross domain evaluation (Table 2). The work could be strengthened by adding more topics from MMLU or at least mentioning how the considered subset has been selected.

**Study over multiple large models**

The method impact is compared over 4 models (Gemma and LLama) which shows that the results are likely to be extendable to other setups. The model size is adequate but reporting results with smaller models to compare with prior work seems necessary (see weaknesses).

**Weaknesses:**

**Missing experiment with topic-agnostic groups**

The paper immediately studies defining the grouping per topic (before then showing that groups from different topics do not overlap much). However, it is not obvious to the reader that pruning in a topic-agnostic manner would not work. E.g. if one defines a single grouping for the whole MMLU, how the result on Figure 6 would look?

**No measurement against the processing speed, resource consumption motivation**

L. 364-368 mentions that the computational benefit is the main motivation of network sparsification. It is necessary to verify that unstructured pruning in the 5%--50% range has a computational benefit. It also seems necessary to verify that the potential gains are compatible with quantization, an established method pursuing the same goal.

**No comparison with alternative strategies**

You mention L39-45 that a contribution of your work is to apply pruning to larger networks than prior research. It seems necessary to run a series of experiments on smaller models to (i) establish the benefit of your approach compared to prior work, (ii) determine if many of the empirical questions on pruning can be answered at small scale and then applied at larger scale. For comparison, I would suggest at least to report 50% sparsity results on MMLU-5 shot with LLAMA-2-7B (and maybe LLAMA-2-13B) to compare with results already in Table 21 of Sun et al 2024.

**Questions:**

**Layerwise Pruning**

Q1: When selecting a group to remove (e.g. group 50%-55%), you remove that group for all matrices in the network? Do you exclude some layers (e.g. the last linear layer)? I imagine that layernorm is untouched, no? Could you specify?

Q2: Related question. When removing a group, does the impact on accuracy vary across layers?


**Section 3. Problem Setup**

Q3: In Equation (1), is the index j in X_j is over the input (1…C_{in})? Or over the output W_{i,j} with W \in R^{C_{in} x C_{out}}? Maybe you meant to define W as a matrix of dimension C_{out} x  C_{in} instead? Please correct this.

**Section 4. Weight Distribution**

Q4: It seems that the definition of the groups you propose correspond to quantiles (Oxford dictionary: any of the groups produced by dividing a frequency distribution into equal groups, e.g. a quartile or percentile). If so please use this common name, if not could you point at the difference?

Q5: The group overlap plot in Figure 2 and 3 seems to indicate that the Philosophy is close to Anatomy but far from Moral and Medicine; it seems counter-intuitive, e.g. Moral and Philosophy are usually dealing with similar topics, far from Anatomy and Medicine which are related. Could you add a comment on that point?

**Section 5. Experiments**

Q6: It seems that ‘Phychology’ (L389, L417) should be spelled ‘Psychology’ no?

Q7: Figure 6 seems to report the pruning impact for 10 of the 20 groups, what are the results on the remaining 10? Could you report them or mention why they are omitted?

Q8: In Table 1, how many instances are used to define the groups? You could mention this explicitly in the text.

Q9: In Table 1, once the groups are defined, how do you select the group(s) to prune? Do you measure the performance on a limited number of validation instances or on the test data? Is there a difference in the groups that would be optimal for either?

Q10: In Table 1, do you always prune a single group (i.e. meaning that e.g. 2% correspond to 1 of 50 groups, while 50% correspond to 1 of 2 groups, etc)? You could mention this explicitly in the text.

Q11: Does Table 2 correspond to 5% pruning? Could you mention it in the caption?

Q12: In Table 1 and Table 2, red arrows denote improvements and green arrows denote deterioration. It is more common to use red for negative outcomes and green for positive outcomes. Could you invert the colors?

Q13: In Figure 7, it is not clear to me what #1, … #10 means here, does #1, (resp #2) correspond to groups named  0-10% (resp 10%-20%) in the rest of the paper? Could you unify the notation?

Q14: In Figure 7, it is difficult to identify the cases which perform better than the full model, could you highlight these? And determine if the improvement is significant, e.g. above the standard deviation of a bagging estimate?

Q15: In Figure 7, is it possible to identify the best performing group on the test data using the validation data?

Q16: Could you provide the numerical results for Figure 6 and Figure 7 in Appendix?

---

### Official Review · Reviewer_xxiV · 2024-11-04

**Soundness:** 3
**Presentation:** 3
**Contribution:** 3
**Rating:** 6
**Confidence:** 3

**Summary:**

* The authors show that it is possible to extract subnetworks from LLMs that specialize in specific domain tasks.
* The proposed probing method is applied to Llama and Gemini models, suggesting their generalizability across different models.
* The subnetworks outperform the full model in their respective domains, but suffer from performance degradation in out-of-domain tasks (as expected).

**Strengths:**

* Detailed analysis of the weight values across model layers and across different models (Llama, Gemini) to justify the weight scoring method and identify common patterns across domains
* Compelling build-up from motivating observations to concrete algorithms for probing and subnetwork discovery
* Evaluation of the full and subnetwork models across different domains and relevant datasets (GSM8K, MMLU, HumanEval)

**Weaknesses:**

* Model variety. While I'm aware there could be resource constraints, I would have liked to see more size variety in the LLMs considered. Llama 7 /13B and Gemini 7 / 9B are in the same weight class, and pruning could arguably be more interesting in the context of even larger models.
* Lack of performance (throughput / latency / memory footprint) analysis. It would have been nice to see what the actual effects of pruning are, other than the fact that masks are employed to zero-out certain weight components.
* Comparison with baselines. The result metrics reported are only for their proposed method; there is no comparison with any previously proposed techniques, so it is hard to gauge how effective the pruning scheme is relative to other methods.

**Questions:**

* What is the intuitive justification behind the weight scoring scheme? This seems somewhat arbitrary / handcrafted. Would using a different scheme yield a different subnetwork, e.g., using L1 instead of the L2 norm?
* Does combining / ensembling discovered subnetworks from similar domains (e.g., code and math) yield any performance gains compared to a single subnetwork?
* Equation (6) has a strong linearity assumption. How is the presence of activations in FFNs accounted for in the algorithm (or is it insignificant enough that we can ignore it)?
* The pruning algorithm goes through the layers sequentially from 1 to L. Does this introduce any bias? If we decide to iterate layers backwards or in random order, how would the subnetwork differ from that identified by pruning sequentially?

---

### Official Review · Reviewer_HPXM · 2024-11-05

**Soundness:** 2
**Presentation:** 2
**Contribution:** 2
**Rating:** 3
**Confidence:** 3

**Summary:**

This paper makes several observations about large language model’s generalization and specialization: (1) weight activation in general-purpose language models are task specific, (2) weights can be pruned for a specific task, and this improves task performance, (3) pruned model has reduced generalization to other unrelated tasks.
Experiment were mostly conducted with Llama-2-7B, while Gemma-7B and Llama-2-13B experiments are done to further verify the universality of these observations.

**Strengths:**

* Paper is mostly written clearly.
* Motivation is clear: prior work focuses on smaller language models and this work conducts the investigation to larger models.

**Weaknesses:**

* Compared to prior work, novelty in methodology or new insights is unclear. In particular, the question associated with motivation is not answered. Do larger models work differently compared to smaller models? What are insights that are new in this work, or inconsistent with prior work? I hope the paper highlight these contributions in a more straightforward way.
* The visualization and captions can be improved for better clarity. It is hard to understand the figures in some cases.
* Claims in the paper (efficiency, performance improvement) are not backed by experiments or other evidence.
* Lack of actionable suggestions given the observations made in the work.

**Questions:**

* Line 165, what does “the pruned model always performs lower than the full model” mean? This seems to be contradicting with the previous sentence which suggests comparable performance.
* Figure 2: How to read this figure? Is 0-5% the group with the highest score, or is 95-100% the one? Does larger radius mean larger overlap? Does the full radius mean an overlap of 100%?
* Figure 3: Why are these layers selected in particular?
* Line 403, “compared to the previous method, … our approach splits weight neurons into groups…” What is the reasoning behind the design of grouping?
* Line 406, “by probing the group weights, we achieve a more refined and efficient elimination of weight neurons” I’m not fully convinced by this argument. Is there any experiment to support this? Running and comparing with the method in Sun et al., 2024 will make this argument more convincing.
* Line 467, “in all models, pruning the weights with low weight scores does not result in optimal subnetworks.” Is this result presented in one of the tables?

__Suggested related work.__
While the paper extensively reviewed prior works, I find the following two papers highly relevant. The paper can be further strengthened by discussing the differences and similarities with these works:
* Task-Specific Skill Localization in Fine-tuned Language Models. Panigrahi et al. (ICML 2023)
* When Parts Are Greater Than Sums: Individual LLM Components Can Outperform Full Models. Chang et al. (EMNLP 2024)

---

### Official Review · Reviewer_tAQy · 2024-11-09

**Soundness:** 2
**Presentation:** 1
**Contribution:** 2
**Rating:** 3
**Confidence:** 4

**Summary:**

The paper presents a way of identifying weights that are less important for a specific domain and then masking them. In particular the importance being used is the one defined by Sun et al (2024), namely absolute value of a weight multiplied by the norm of the features (features that are input to this weight). Subsequently, the weights are grouped by sorting them and choosing the N first ranks the N second and so on and so forth. Whole groups are masked as one in the proposed algorithm and then the performance is checked against a validation set and the masking is only allowed if it leads to an improvement. The authors show that this masking can result in improvements for in-domain accuracy while reducing the overall accuracy. It is also claimed as a method for explaining the generalization of such models.

**Strengths:**

- Weight pruning is an important and active area of research in large models and specifically language models.
- Similarly identifying smaller subnetworks without retraining can be really powerful in terms of deploying large models or personalization.
- The analysis of "weight overlap" on different domains could have been interesting if better presented.
- It is interesting that doing a model search with masking can result in improved performance and possibly made more efficient using the proposed grouping.

**Weaknesses:**

- The most important weakness of the paper is that there is no clear goal or conclusion from any of the proposed methods or experiments.
  - The analysis based on the weight overlap is not clearly presented, different layers and different models for every plot with minimal explanation mostly focusing on describing what is shown without providing any insight that may stem from the analysis itself. For example, in figure 3, Gemma 7B has more overlap with different domains than Gemma-2 9B has with the same domain. Similarly, for math, code and medicine domains, the behavior is extremely different with minimal overlap (compared to figures 2 and 3) even within the same domain.
  - The analysis in subnetwork probing seems flawed. It is interesting that doing a search with masking actually results in improved performance, it is also interesting that it can be more efficient compared to random masking or single weight masking. However, it should be treated and compared as such, a search in the masking space. Removing the least important weights does not result in better performance which means that the method relies on lines 10-12 of the algorithm. As a result the method should be compared to gradient based search methods in a FLOP equalized manner. Algorithm 1 requires G evaluations of the validation set and G*L forward passes on the training set so it is not an insignificant FLOP investment.
- The paper is poorly written with a lot of typos.

**Questions:**

The main thing to improve the paper in my opinion would be to clarify its goals. The authors show that the weight sets grouped by importance may be a useful tool but the experiments do not show how.
1. If even the least important weights cannot be removed then are the plots of figures 2-5 useful?
2. If the search space is useful shouldn't the method be compared to gradient based methods?

---

### Note · Authors · 2024-11-25

I have read and agree with the venue's withdrawal policy on behalf of myself and my co-authors.